# Recent Advances in Surface Modifications of Elemental Two-Dimensional Materials: Structures, Properties, and Applications

**DOI:** 10.3390/molecules28010200

**Published:** 2022-12-26

**Authors:** Junbo Chen, Chenhui Wang, Hao Li, Xin Xu, Jiangang Yang, Zhe Huo, Lixia Wang, Weifeng Zhang, Xudong Xiao, Yaping Ma

**Affiliations:** 1Key Laboratory of Quantum Matt Science, Henan Key Laboratory of Photovoltaic Materials, Henan University, Zhengzhou 450046, China; 2School of Physical Science and Technology, Wuhan University, Wuhan 430072, China; 3Key Laboratory of Artificial Micro- and Nano-Structures of Ministry of Education, Wuhan University, Wuhan 430072, China; 4State Key Lab of Optoelectronic Materials and Technologies, Guangdong Province Key Laboratory of Display Material and Technology, School of Electronics and Information Technology, Sun Yat-sen University, Guangzhou 510275, China

**Keywords:** elemental two-dimensional materials, allotropic structures, surface modifications, properties, applications

## Abstract

The advent of graphene opens up the research into two-dimensional (2D) materials, which are considered revolutionary materials. Due to its unique geometric structure, graphene exhibits a series of exotic physical and chemical properties. In addition, single-element-based 2D materials (Xenes) have garnered tremendous interest. At present, 16 kinds of Xenes (silicene, borophene, germanene, phosphorene, tellurene, etc.) have been explored, mainly distributed in the third, fourth, fifth, and sixth main groups. The current methods to prepare monolayers or few-layer 2D materials include epitaxy growth, mechanical exfoliation, and liquid phase exfoliation. Although two Xenes (aluminene and indiene) have not been synthesized due to the limitations of synthetic methods and the stability of Xenes, other Xenes have been successfully created via elaborate artificial design and synthesis. Focusing on elemental 2D materials, this review mainly summarizes the recently reported work about tuning the electronic, optical, mechanical, and chemical properties of Xenes via surface modifications, achieved using controllable approaches (doping, adsorption, strain, intercalation, phase transition, etc.) to broaden their applications in various fields, including spintronics, electronics, optoelectronics, superconducting, photovoltaics, sensors, catalysis, and biomedicines. These advances in the surface modification of Xenes have laid a theoretical and experimental foundation for the development of 2D materials and their practical applications in diverse fields.

## 1. Introduction

Graphene, initially isolated from graphite via mechanical exfoliation, possesses a layered honeycomb structure and exhibits fascinating electrical and thermal properties [1]. Subsequently, two-dimensional (2D) materials, including elemental monolayers (Xenes) [2], transition metal chalcogenides (TMCs) [3,4], oxides [5], halides [6,7], and carbides (MXenes) [8,9], have shown intriguing properties, such as high carrier mobility [10], layer-dependent band structures and magnetic properties [6,11,12], nontrivial topology [13,14,15,16], valleytronics [17], etc. Therefore, 2D materials have become promising candidates for various applications relating to next-generation technology, including spintronics, superconducting, nanoelectronics, nanosensing, etc.

Over the past few years, many studies have focused on searching for other 2D Xenes with distinctive and exciting properties beyond graphene (Figure 1a) [18,19]. The successful experimental realization of non-graphene 2D analogs (silicene and phosphorene) sparked a continuous expansion of the list of elements with atomically thin forms [20,21]. Sixteen elemental-main-group 2D Xenes have been predicted theoretically or created experimentally to date (Figure 1b) [22,23,24,25,26,27,28,29,30,31,32]. To the best of our knowledge, except aluminene and indiene, the other 13 non-graphene Xenes have been experimentally obtained (Figure 1c). Currently, monolayer or few-layer Xenes can be created using mechanical exfoliation, liquid phase exfoliation, and epitaxial growth methods [1,12,22,33].

Even though 2D Xenes have demonstrated great potential in numerous applications, their intrinsic properties usually constrain the expansion of these applications. For instance, despite the high mobility, the gapless band structure in graphene restricts its applications in electronic devices. Therefore, finetuning the properties of 2D Xenes plays an essential role in overcoming their intrinsic constraints [34,35,36,37]. Surface modifications can effectively tailor the properties of 2D Xenes for practical applications. The common approaches for surface modifications include heteroatom-doping [38,39], adsorption [34,36,40], in-plane heterostructure formation [41,42,43], boundary formation [44], edge shape [32,45,46,47,48], phase transition [49], strain [50,51], twisted structure [52,53], etc. Motivated by recent advances in the surface modifications of 2D Xenes, herein, we review the reported work to date, connect the surface modifications of various 2D Xenes to their measured/predicted properties, and evaluate their advantages and disadvantages for various applications.

## 2. Classification of 2D Xenes

The material properties (electronic, optical, chemical, etc.) of 2D Xenes are not only determined by their chemical compositions but are also strongly associated with the atom arrangement in the lattice. Due to the preferred orbital hybridization of various elements in the main group, 2D Xenes have been theoretically predicted or experimentally verified to possess allotropes with diverse crystal lattices (Figure 2). The reported 16 Xenes are classified into main groups, including group III (borophene, aluminene, gallenene, indiene, and thallene); group IV (graphene, silicene, germanene, stanene, and plumbene); group V (phosphorene, arsenene, antimonene, and bismuthene); and group VI (selenene, and tellurene).

In group III elements, theoretical calculations have predicted many allotropes of 2D borophene (B_1−ν_☐_ν_; ν represents the vacancy concentration) with various vacancy concentrations and 2D aluminene with various forms [55,59,60,61]. However, only two allotropes of 2D gallenene (buckled and planar structures) have been successfully exfoliated [29]. Similarly, 2D indiene has been predicted to possess buckled, planar, and puckered geometry [62]. Among group IV elements, the favorable hybridization state is somewhere between sp^2^ and sp^3^, leading to various 2D allotropic structures, such as planar honeycomb graphene/stanene [63,64,65], buckled silicene/germanene/stanene/plumbene [25,30,66,67], pha-/penta-graphene [68,69], MoS_2_-like stanene [57], and honeycomb dumbbell (HD)/large honeycomb dumbbell (LHD) silicene/germanene [57]. Different from group III and group IV, all the elements in group V prefer the sp^3^ hybridization state to create buckled (α-form) or puckered (β-form) lattices [24,70,71,72,73,74,75]. Meanwhile, a γ-form and a δ-form of arsenene have also been proposed [56]. In group VI, four different allotropic forms of tellurene have been predicted [27,58], while Se can form a chain, a ring, or a square structure [58]. To verify the atomic structures of various elemental Xenes, epitaxial growth has been attempted on many substrates, analogous to the epitaxial growth of graphene. The suitable substrates for the epitaxial growth of various non-graphene Xenes (Figure 3) reveal that the substrate choices affect whether the synthesis can succeed in addition to an allotropic lattice structure.

## 3. Surface Modifications of 2D Xenes in Various Main Groups

### 3.1. Group III

**Borophene.** Borophene, a monolayer of boron atoms, has been synthesized on many metal substrates under ultrahigh vacuum conditions [22,76,77,78]. Several phases of borophenes, including 2-*Pmmn*, ν_1/6_ (β_12_), ν_1/5_ (χ^3^), ν_1/12_, and honeycomb, which are realized experimentally, exhibit metallic properties [22,76,77,78]. However, plenty of borophene allotropes with various vacancy concentrations have been reported with theoretical calculations [59,60,79,80,81]. Moreover, the relationship between the formation energy and vacancy concentration shows a V-shaped function [82], demonstrating that the vacancy concentration is directly correlated with the stability of borophene.

Borophene possesses some unique physical and chemical properties. Due to the strong B-B bonds and distinctive atomic structure, borophene exhibits an ultrahigh mechanical modulus [83,84]. Young’s modulus of the *Pmmm* and 2-*Pmmn* phases of borophene along the armchair direction can reach 574.61 and 398 N m^−1^, respectively, larger than that of graphene. In fact, several factors, including the vacancy concentration, chemical modification, layer numbers, and temperature, can affect the mechanical properties of borophene. Young’s modulus of borophene was found to decrease with an increasing vacancy concentration, layer numbers, and temperature, as well as hydrogenation or fluorination [83,84,85,86,87,88,89]. Because of its outstanding flexibility and excellent electronic conductivity, borophene has a wide range of intriguing applications in flexible electronic devices [79]. In addition, the thermal conductivity of the 2-*Pmmn* phase of borophene is different along the zigzag and armchair directions due to its highly anisotropic atomic structure [90,91]. The electronic band structures of borophene show 1D, nearly free electron states and metallic Dirac fermions [92,93,94,95]. However, the metallic-to-semiconducting transition can be achieved via fluorination and a uniaxial or biaxial strain [85,96,97]. Last but not least, the superconductivity of borophene is the most notable characteristic, sparking plenty of research interest [80,98,99,100,101,102,103,104,105,106]. The superconducting transition temperatures (T_c_) can be tuned with vacancy concentration, doping, strain, and Mg intercalation. T_c_ exhibits a V-shaped function as the hexagon hole density [99], illustrating that T_c_ gradually decreases with a rising boron vacancy concentration of up to ν = 1/9; thereafter, T_c_ steadily increases with the vacancy concentration. Furthermore, tensile strain or hole-doping can increase T_c_; in contrast, a compressive strain or electron-doping decreases T_c_ [106]. The suppression induced by electron-doping makes it difficult to experimentally probe superconductivity in substrate-supported borophene.

The physical and chemical properties of borophene can be tuned using surface modifications, such as hydrogenation, fluorination, doping, intercalation, strain, etc. Young’s modulus of borophene decreases after hydrogenation or fluorination. Furthermore, hydrogenation can lead to Dirac cones with massless Dirac fermions in *C2/m*, *Pbcm*, and *Pmmn* structures, while *Cmmm* structures exhibit Dirac ring features (Figure 4a–d) [107]. Interestingly, BH sheets have been successfully prepared from MgB_2_ by using the cation exchange method (Figure 4e,f) [108]. For Mg intercalation, the intercalated bilayer borophene (B_2_MgB_2_) can exhibit good phonon-mediated superconductivity with a high T_c_ of 23.2 K (Figure 4g,h) [102]. Moreover, tensile strain in borophene is beneficial for superconducting [106]. Li-doped borophene–graphene heterostructure shows gas-sensitive properties, and this is promising for borophene-based gas sensors [109].

**Gallenene and thallene.** The surface modifications of aluminene and indiene have not been reported before, as studies on them are still in the theoretical research stage, without experimental realization. Therefore, this section will focus on gallenene and thallene. In 2018, few-layer gallenene was first obtained using the solid melt exfoliation technique [29]. Thereafter, the epitaxial growth method was used to prepare gallenene. The substrate and the loading amount of gallium can modify the atomic and electronic structures of gallenene. With a low loading amount of gallium, the monolayer gallenene grown on Si(111) displays a 4 × √13 superstructure (Figure 5a–d), while the second-layer gallenene exhibits a hexagonal honeycomb structure with a high loading amount (Figure 5e,f) [110]. The buckled honeycomb gallenene shows metallic properties (Figure 5g) [110]. Nevertheless, the growth behavior of gallenene on a GaN(0001) substrate is totally different, showing a bilayer flat gallenene (Figure 5h,i) [111]. Excitingly, the bilayer hexagonal gallenene exhibits superconducting with a T_c_ of 5.4 K (Figure 5j,k) [111]. However, thallene has rarely been reported. Recently, honeycomb thallene was successfully formed on a NiSi_2_/Si(111) substrate [112].

### 3.2. Group IV

**Graphene.** The sp^2^ hybridization of carbon atoms leads to the formation of flat honeycomb graphene with a σ bond between neighboring carbon atoms [113]. The σ bond is the key to the high robustness of graphene, with a Young’s modulus of 1T Pa and a fracture strength of 130 GPa [114]. The unhybridized p orbit, perpendicular to the planar structure of graphene, binds covalently with neighboring carbon atoms to form a π band that is half-filled. Graphene is a semimetal, showing linear dispersion bands near the Fermi level with massless Dirac fermions [113]. As a result, graphene displays an ambipolar electric field effect and high carrier mobility [115]. In addition to its excellent transparent properties, graphene becomes a low-cost alternative to indium tin oxides [116]. Furthermore, graphene exhibits impressive thermal properties with a thermal conductivity ranging from 3000–5000 W m^−1^ K^−1^ [117].

Graphene’s distinct physical characteristics make it potentially useful for field-effect transistors (FET), sensors, transparent conductive films, energy devices, etc., but its intrinsic gapless character still constrains its further applications. First, heteroatom-doping or chemical adsorption can effectively tune the electronic properties of graphene. N-doping can not only induce an n-type-doping effect (Figure 6a–c) [118] but can also give rise to p-type-doping with different configurations of N substitutions. For example, graphitic N induces n-type-doping, but pyridinic N induces p-type-doping [119]. In addition, B-doping can introduce p-type transfer characteristics [120]. Chemical functional groups can also produce various doping effects. For example, the adsorption of spiropyran and DR1P molecules introduce n-type and p-type-doping to graphene, respectively [36,121]. Moreover, light can reversibly switch the molecular transformations, resulting in the controllable shift of the Dirac point of graphene (Figure 6d) [36,121]. By seamlessly and precisely stitching the domains of graphene and h-BN (Figure 6e), the hybrid atomic layers of in-plane heterostructures can be applied for intriguing electronic applications [42,122]. Interestingly, in the proper twisted angles of bilayer graphene, the electronic band structure shows flat bands near Fermi energy, resulting in the correlated insulating states at half-filling and unconventional superconductivity with a T_c_ of 1.7 K (Figure 6f,g) [52].

**Silicene.** Silicene, the silicon analog of graphene, was first predicted in a theoretical study [123] and first realized experimentally via epitaxial growth on Ag(110) [20]. Unlike the way the sp^2^ hybridization of carbon atoms induces a flat honeycomb structure in graphene, silicon prefers mixed sp^2^-sp^3^ hybridization to form low-buckled honeycomb silicene, retaining the existence of Dirac fermions [124,125]. Considering the spin-orbit coupling (SOC) effect, silicene is predicted to have a spin-orbit gap of 1.55 meV, much larger than that of graphene [16]. Owing to its topologically nontrivial electronic structures, silicene exhibits many unique physical properties, including the quantum spin Hall (QSH) effect [16,126], giant magnetoresistance [127,128], the field-tunable bandgap [129,130], and nonlinear electro-optic effects [131]. Hence, silicene shows great potential for device applications, especially for gate-controlled topological FET [132]. Although silicene has been prepared on many substrates, the poor air stability of silicene is the major challenge, requiring the proper encapsulation or passivation of the reactive surface for device fabrication [133,134].

Due to the limitations of the poor air stability of silicene, it is difficult to experimentally perform surface modifications on silicene. Theoretical investigations on the surface modifications of silicene focus on doping, strain, hydrogenation, intercalation, and chemical adsorption. Transition metal adsorption can induce various doping effects in silicene (n-type via Cu, Ag, and Au adsorption; p-type via Ir adsorption; and neutral type via Pt adsorption in Figure 7a) [135], and so can applying strain [136]. Moreover, Mn-doping can induce a ferromagnetic state in silicene, which can be transformed into an antiferromagnetic state with the application of biaxial strain (Figure 7b) [137]. Under a certain level of pressure strain, the spin-orbit bandgap of silicene will increase from 1.55 to 2.9 meV [16]. One-side semi-hydrogenation can introduce ferromagnetism to silicene, as well as make it semiconducting with a direct bandgap of 1.74 eV (Figure 7c,d) [138]. Oxygen intercalation into underlying silicene on a Ag(111) surface can effectively reduce the orbital hybridization of the top-layer silicene and Ag substrate, leading to massless Dirac fermions (Figure 7e–g) [139]. However, in K-intercalated bilayer silicene, the Dirac cones are recovered with a small bandgap of 0.27 eV [140]. Chemical adsorption (gas and organic molecules) can tune the electronic properties of silicene, which could be a better candidate for detecting gas and organic molecules compared with graphene [109,141,142].

**Germanene.** Germanene, similar to silicene, shows a low-buckled honeycomb structure, leading to a topologically nontrivial electronic structure with a spin-orbit bandgap of 23.9 meV due to a greater SOC than that of graphene and silicene [16,143,144]. These characteristics make germanene a promising candidate for applications in high-speed and low-energy-consumption devices since it has high charge-carrier mobility and exhibits QSHE [144]. Germanene has also been successfully prepared via epitaxial growth on various substrates.

Doping various atoms can introduce totally different influences on the physical properties of germanene. For instance, the adsorption of alkali metal atoms makes semi-metallic germanene become metallic, with the Dirac point moving below the Fermi level and an opened small bandgap at the Dirac point, while the adsorption of halogen atoms could lead to relatively large bandgaps, ranging from 0.416 to 1.596 eV, which is promising for optoelectronic applications [145,146,147]. The adsorption of transition metal atoms (e.g., Ti, Sc, V, Cr, Mn, Fe, and Co) can induce magnetism, while nonmagnetic semiconducting states can be realized for Ni, Cu, and Zn adsorption [148,149]. The atomic structures of germanene can be controlled with supported substrate and growth conditions. Directly grown on a Ag(111) surface, two distinct phases of germanene can be observed: one shows a striped phase, a honeycomb lattice partially commensurate with the substrate; the other is a quasi-freestanding phase, a honeycomb lattice incommensurate with the substrate [150]. By epitaxially preparing it on Ag(111) thin-film grown on Ge(111) with a segregation method, the germanene shows a highly ordered long-range superstructure with two types of protrusions (hexagon and line), resulting in a (7√7 × 7√7)R19.1° supercell with respect to Ag(111) (Figure 8a,b) [151]. However, the single domain (3 × 3) and multidomain (√7 × √7)R(±19.1°) of germanene can exist simultaneously on an Al(111) surface (Figure 8c) [152]. On MoS_2_ substrate, germanene islands can be formed at high deposition rates, whereas Ge atoms prefer to intercalate between MoS_2_ layers to form Ge clusters at low deposition rates [147,153]. On a Au(111) surface, honeycomb (1 × 1) germanene with a buckled structure was identified in a (√7 × √7) superstructure, exhibiting distinctive vibrational phonon modes and enhancing electron–phonon coupling induced by the tensile strain [154].

**Stanene.** Analogously, monolayer Sn prefers to form a low-buckled structure due to the mixed sp^2^-sp^3^ hybridization of Sn atoms [155]. Stanene has been predicted to have massless Dirac fermions and open a spin-obit bandgap of 0.1 eV at the K point with SOC [155]. The bandgap at the Dirac point can produce a conductive 1D helical edge state with opposite spin polarization, allowing for low-power spintronic applications [13,156].

Double-side-decorated stanene created by chemical functional groups appears in the most stable configuration (Figure 9a). For pristine stanene, SOC can open a bandgap of 0.1 eV at the K point; thus, stanene becomes a QSH insulator (Figure 9c). After hydorgenation or fluorination, the bandgap at the K point is substantially enlarged because of the saturation of the π orbital (Figure 9d,e). Fluorination induces a parity exchange between the occupied and unoccupied bands at the Γ point. In detail, a negative-parity Bloch state shifts downward into valence bands, leaving a positive-parity Bloch state as the conduction band minimum (Figure 9d), leading to a topologically nontrivial band structure. Nonetheless, the band inversion at the Γ point cannot be seen in hydrogenated stanene (Figure 9e). In fact, the band inversion exists for several chemical functional groups (halogen atoms and -OH) (Figure 9b).

Stanene is predicted to show superior sensing performance for small molecules. CO, O_2_, NO, NO_2_, and SO_2_ molecules act as charge acceptors, whereas H_2_O, NH_3_, and H_2_S molecules act as charge donors [157,158]. In addition, molecule adsorption can effectively tune the work function of stanene (Figure 9f) [157]. The edge shapes of stanene play a key role in the physical properties of stanene nanoribbons (NRs). Armchair stanene NRs are nonmagnetic semiconductors with tunable bandgaps by ribbon width, whereas zigzag stanene NRs present antiferromagnetic ground states with an opposite spin order between the two edges [159]. Generally, the energy gap (0.1 eV) of monolayer stanene rules out phonon-mediated superconductivity. Interestingly, doping with Ca (Li) can lead to superconductivity with a low T_c_ of ~1.3 K (~1.4 K), lower than the value (3.7 K) of bulk β-tin [160].

**Plumbene.** Unlike the other four Xenes in main group IV, no Dirac point crosses linearly from the Pb p_z_ orbit at the K point without SOC. Although turning on SOC opens a large bandgap of ~400 meV, no Dirac edge state has been observed in the bandgap of plumbene [161]. In addition, resulting from the energy decrease in the s antibonding state from graphene to plumbene, the s antibonding state is lower than all p bonding and antibonding states at the Γ point in plumbene, totally different from graphene, silicene, germanene, and stanene [162]. Therefore, plumbene is a normal insulator with a topologically trivial character. However, through electron-doping, plumbene can become a topological insulator with a large bulk gap (~200 meV) [163]. Plumbene decorated by chemical function groups (hydrogen and halogen atoms) can transform from a normal insulator to a QSH insulator with giant bulk gaps from 1.03 to 1.34 eV [161]. Plumbene has been predicted to be magnetic with Ti-, V-, Cr-, Mn-, Fe-, and Co-doping, while Sc- and Ni-doped plumbene is nonmagnetic [164]. It is interesting that plumbene can be successfully grown using segregation on a Pd_1−x_Pb_x_(111) alloy surface [30]. Furthermore, a c(2 × 4) structure of Pb forms on Ir(111) substrate, whereas a flat honeycomb plumbene can be formed on an Fe monolayer on Ir(111) [165].

### 3.3. Group V

**Phosphorene.** Puckered and buckled structures in phosphorene are the most common allotropic monolayer structures, corresponding to the individual atomic layers of black phosphorous and blue phosphorous crystals, respectively [166]. Puckered monolayer phosphorene is semiconducting with a direct bandgap of 1.83 eV, whereas buckled monolayer phosphorene is a semiconductor with an indirect bandgap of 2.0 eV. Phosphorene can be obtained through mechanical exfoliation, liquid phase exfoliation, electrochemical exfoliation, chemical vapor deposition, epitaxial growth, etc. [166]. To date, the semiconducting character of phosphorene has led to some experimental demonstrations in various applications, including electronics, optoelectronics, photovoltaics, supercapacitors, and catalysis [73,167].

Transition metal-doped black phosphorene possesses dilute magnetic semiconducting properties. In particular, the substitutional doping of Ti, Cr, and Mn can create a spin-polarized semiconducting state, while a half-metallic state is realized via V- and Fe-doping (Figure 10a) [168]. Both Fe-doping and N-doping can significantly improve the electrocatalytic activity of black phosphorene for nitrogen reduction reactions [169,170]. For blue phosphorene, B-doping and C-doping can both improve the sensitivity of NH_3_ gas molecules, and the sensitivity of CO gas molecules can be enhanced by B-doping [171]. With a monotonic increase in an external electric field, black phosphorene can transition from a normal insulator to a topological insulator and, eventually, to a metal (Figure 10c–e) [172]. In addition, an external electric field can realize reversible potassium intercalation in a blue phosphorene–Au network (Figure 10b) [173]. When axial strain is applied in the zigzag or armchair direction, the bandgap of black phosphorene will exhibit a direct–indirect–direct transition (Figure 10f,g) [51]. Moreover, a topological phase transition of black phosphorene can be realized via the application of compressive biaxial in-plane strain and perpendicular tensile strain [174].

**Arsenene.** Arsenene mainly has two kinds of allotropic structures, including buckled and puckered [166]. Buckled honeycomb monolayer arsenene, derived from semi-metallic gray arsenic, has an indirect bandgap of 2.49 eV [24], while puckered monolayer arsenene, exfoliated from black semiconducting arsenic, possesses an indirect bandgap of 0.831 eV [75]. Both buckled arsenene and puckered arsenene have thickness-dependent bandgaps. The methods to obtain arsenene include top-down (mechanical exfoliation and liquid phase exfoliation) and bottom-up strategies (molecular beam epitaxy, chemical vapor deposition, physical vapor deposition, etc.) [175]. For epitaxial growth, monolayer buckled arsenene was successfully grown on a Ag(111) substrate [176], whereas monolayer armchair arsenene nanochains have been formed on a Au(111) surface [32]. Two-dimensional arsenene has been theoretically predicted to exhibit various physical properties, such as an indirect-to-direct bandgap transition, a semimetal-to-semiconductor transition, superconductivity, and a QSH effect, deserving many research efforts [24,177,178]. Recently, few-layer black arsenene was proved to exhibit a particle–hole asymmetric Rashba valley and exotic quantum Hall states due to synergetic effects between the spin-orbit interaction and the Stark effect [179].

Substrate temperatures can modify the formation of arsenic nanostructures. On Au(111) substrate, the arsenic monolayer formed by 0D tetrahedral As_4_ clusters can transform into a monolayer formed by 1D armchair arsenene nanochains (Figure 11a) [32]. The application of biaxial tensile strain can effectively tune the band structures of arsenene. Under low tensile strain, arsenene can transform from an indirect to a direct bandgap (Figure 11b,c) [24]. By further enlarging the tensile strain, the direct bandgap will gradually disappear, causing band inversion at the Γ point (Figure 11d,e) [177]. The consideration of SOC can open a spin-orbit bandgap (~131 meV) under a tensile strain of 18.4% (Figure 11f), indicating a QSH effect in arsenene [177]. Intriguingly, under proper biaxial tensile strain and electron-doping, arsenene can be superconducting, with a T_c_ of 30.8 K [178]. Indeed, 3D-transition-metal-doping can strongly tailor the electronic and magnetic properties of arsenene. Ti-, V-, Cr-, Mn-, and Fe-doping can induce magnetic states for arsenene [180]. Meanwhile, Ti- and Mn-doping leads to a half-metallic state, while V-, Cr-, and Fe-doping results in a spin-polarized semiconducting state [180]. In addition, doping can further modify chemical properties, making arsenene potentially useful for hydrogen evolution and oxygen evolution reactions (Figure 11g) [181].

**Antimonene and bismuthene.** The most stable structures of antimonene and bismuthene are α-form (puckered) and β-form (buckled), which are the monolayers of black and gray bulk allotropes, respectively. Monolayer β-Sb and β-Bi possess indirect bandgaps of 2.28 and 0.99 eV, respectively, whereas the direct bandgaps of monolayer α-Sb and α-Bi are 1.18 and 0.36 eV, respectively [74,182,183]. Notably, the calculated bandgaps may vary depending on the methods used. Both antimonene and bismuthene exhibit tunable bandgaps, high carrier mobility, high stability, and in-plane anisotropy, providing a basis for multifunctional applications in electronics, optoelectronics, sensors, batteries, etc. [184,185,186].

Doping with 3d transition metal atoms for antimonene can lead to significant changes in the bandgap and the magnetic moment [187]. For Cr-doped β-antimonene, a spin-polarized semiconducting state was predicted. For Ti-, Mn-, and V-doped β-antimonene, half-metallic behavior was calculated. Similarly, a Cr-doped bismuthene structure leads to a spin-polarized semiconducting state, while V-doped bismuthene can produce a magnetic metal character, and Mn- and Fe-replacing systems result in half-metal features [188]. Additionally, V-doped systems exhibit ferromagnetism (FM) when two V atoms are far apart, but Cr-, Mn-, and Fe-doped bismuthene exhibits anti-ferromagnetism (AFM) when two impurity atoms are close together or far apart [188]. Bivacancy-doping in β-antimonene can reduce the bandgap of pristine β-antimonene, but monovacancy-doped β-antimonene exhibits a metallic character [189]. Electron-doping and Ca-intercalation can transform bilayer β-antimonene from a semimetal into a superconductor [190]. Moreover, the physisorption of the organic molecules tetrathiafulvalene and tetracyanoquinodimethane can induce n-type- and p-type-doping for antimonene, respectively [191]. Under a monotonic increase in biaxial tensile strain, β-antimonene and β-bismuthene undergo indirect-to-direct bandgap and semiconducting to semi-metallic transitions and even topological phase transitions [24,192,193].

### 3.4. Group VI

**Selenene and tellurene.** Selenene has been predicted to have three allotropic structures, including a 1T-MoS_2_-like structure (t-Se or α-Se), a tiled helical-chain structure (c-Se), and s square structure (s-Se) [194]. Both t-Se and c-Se are semiconductors with indirect bandgaps of 0.71 and 1.74 eV, respectively, while s-Se is semi-metallic. The formation energy of c-Se is the lowest, indicating that c-Se is the most stable phase. In addition, a ring structure of selenene (r-Se) is proposed [58]. However, theoretical investigations predict the existence of three phases of tellurene, including α-, β-, and γ-phases, possessing 1T-MoS_2_-like, rectangle, and 2H-MoS_2_-like structures, respectively [27]. It was found that α- and β-Te show semiconducting characteristics with indirect bandgaps of 0.76 and 1.17 eV, respectively, whereas γ-Te is a metal. It is interesting that the band structures of square selenene and tellurene (s-Se and s-Te) show Dirac-cone-like dispersions. A large bandgap (~0.1 eV) opened by SOC makes them become topological insulators and host nontrivial edge states [58]. Therefore, square selenene and tellurene become promising candidates for spintronic applications. The tensile strain can monotonically decrease the bandgaps of α-Se and α-Te [195,196]. Meanwhile, for 2D square tellurene under a strain effect, the system displays three structural phases, buckled square, buckled rectangle, and planar square phases, which exhibit extraordinary topological properties [196]. In particular, the buckled rectangle tellurene can act as a QSH insulator with a bandgap of 0.24 eV.

## 4. Applications of 2D Xenes

As mentioned above, 2D Xenes possess various physical and chemical properties, such as flexibility, layer-dependent semiconducting, high carrier mobility, molecule and light sensitivity, topologically nontrivial band structures, etc.

In order to utilize 2D Xenes more effectively, surface modifications become particularly important (Figure 12) to tune the properties of 2D Xenes. For instance, the FET of acene-type graphene nanoribbons exhibits excellent semiconductor characteristics with an on/off ratio of 88 [197]. To enhance the mid-infrared (MIR) absorption of graphene, the localized surface plasmon resonance of B-doped Si quantum dots (QDs) results in a QD/graphene hybrid photodetector with ultrahigh responsivity, gain, and specific detectivity in the UV-to-MIR region [34]. A 2D bismuthene/Si(111) heterostructure exhibits excellent photodetection performance in the Vis-MIR region due to the promoted generation and transportation of charge carriers in the heterojunction [198]. Solution-exfoliated black phosphorene flakes, as an electron transport layer, can enhance the performance of organic solar cells [199]. In addition, layered black phosphorene exhibits the selective detection of methanol [200]. The thermoelectric power (S) in black phosphorene can be effectively controlled with ion-gating. In the hole-depleted state, the S of black phosphorene can reach +510 μV/K at 210 K, much higher than the bulk single crystal value of +340 μV/K at 300 K [201]. Under the proper electron-doping and biaxial tensile strain, buckled arsenene shows superconductivity with a high T_c_ of 30.8 K [178]. Iodine-decorated stanene exhibits a topologically nontrivial band structure with a larger gap of ~320 meV than that of pristine stanene (~100 meV) [155]. Graphene/Pt(111) surfaces can cause CO adsorption/desorption and CO oxidation surface reactions [202]. MoS_2_/graphene hybrids decorated by CdS nanocrystals can act as high-performance photocatalysts for H_2_ evolution under visible light irradiation [203]. Moreover, 2D Xenes are also regarded as promising agents for biomedical applications [204]. For example, an ultrathin bismuthene can act as a sensing platform to detect microRNA with a detection limit of 60 PM [205]. Polyethylene-coated antimonene quantum dots can be used as photothermal agents with a high photothermal conversion efficacy of 45.5% for photothermal therapy in cancer theranostics [206]. Overall, thanks to surface modifications, 2D Xenes show great potential for applications in plenty of fields.

## 5. Conclusions

In total, except graphene, 15 different elemental 2D materials of the main group elements have been experimentally created or theoretically predicted to date. In fact, 14 Xenes, including borophene, graphene, silicene, phosphorene, gallenene, germanene, arsenene, selenene, stanene, antimonene, tellurene, thallene, plumbene, and bismuthene, have been successfully grown on proper substrates using epitaxial methods. Although a lot of research, engineering, and development related to 2D Xenes—most of which were investigated theoretically—has been reported in recent years, experimental studies are still desperately required for the development of synthesis strategies and novel applications. Before realizing the surface modifications of 2D Xenes for the applications of interest, three significant challenges need to be overcome:

(i) The synthesis strategies of 2D Xenes must not only ensure reliably large-scale production but also be tailored for the requirements of the application. For example, the applications of electronics and batteries demand low costs and scalable techniques, while plasmonics and spintronics require high fidelity and reproducible techniques. Hence, reliable synthesis approaches play a crucial role in functional design and practical applications.

(ii) The strategies to enhance the environmental stability and mitigate the degradation of 2D Xenes must be taken into consideration for certain applications. For instance, an optoelectronic device based on black phosphorene must be concerned with ambient stability, requiring appropriate encapsulation or the passivation of the surface.

(iii) Many of the predicted and fascinating properties of 2D Xenes require further efforts to find strategies for their implementation, which may offer opportunities to discover revolutionary technologies. In addition, the exciting and novel properties necessitate not just “one-off” research, but also statistical evaluation to ascertain their viability and accessibility on a commercial scale.

Despite the challenges ahead, the exceptional properties of 2D Xenes will significantly impact future applications in various fields. It is hoped that this review will inspire more exhilarating discoveries and applications in the growing family of 2D Xenes.

## Figures and Tables

**Figure 1 molecules-28-00200-f001:**
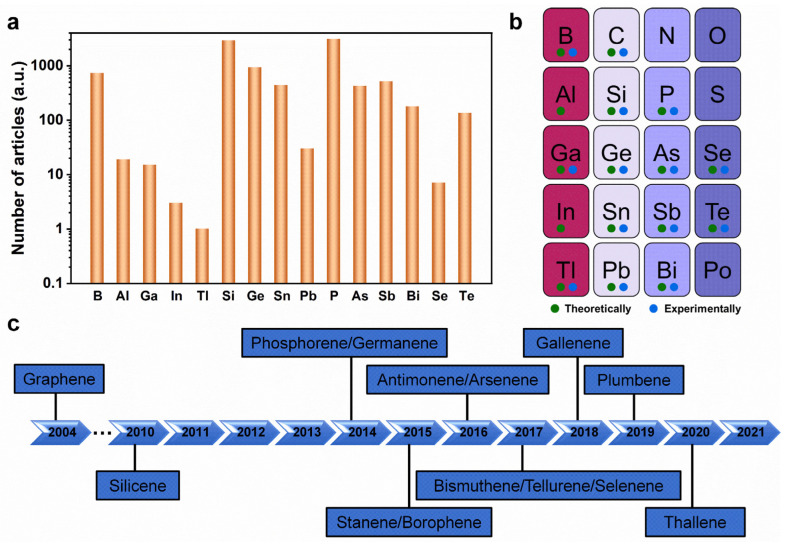
Development of 2D Xenes. (**a**) Statistics diagram of the number of research articles on non-graphene 2D Xenes from 2010 to 2021. (**b**) Overview of 2D analogs of main-group elements, explored using either experimental or theoretical routes. The elements with no signs have not been explored to date. (**c**) The timeline of the experimental creation of 2D Xenes.

**Figure 2 molecules-28-00200-f002:**
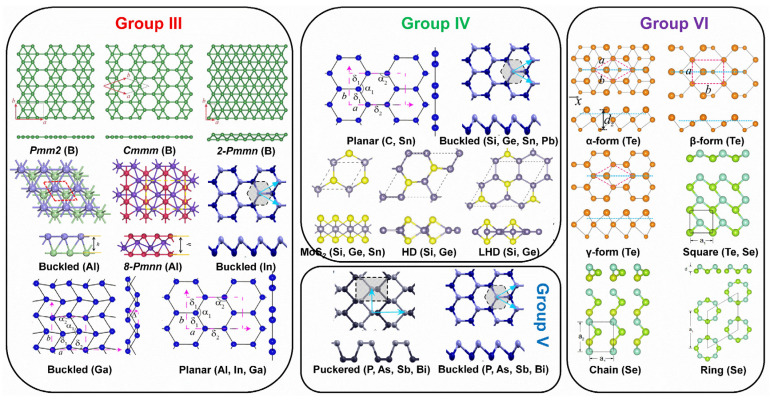
Allotropes of various 2D Xenes in main groups predicted theoretically or created experimentally. The dotted boxes represent the unit cells. Allotropes of borophene: Reprinted with permission from Ref. [54]. Copyright 2017, Informa UK Limited, trading as Taylor & Francis Group. Allotropes of aluminene: Reprinted with permission from Ref. [55]. Copyright 2017, Elsevier Ltd. Allotropes of gallenene: Reprinted with permission from Ref. [29]. Copyright 2018, American Association for the Advancement of Science. Buckled and puckered structures: Reprinted with permission from Ref. [56]. Copyright 2016, American Physical Society. MoS_2_-like, HD, and LHD structures: Reprinted with permission from Ref. [57]. Copyright 2015, American Physical Society. α-form, β-form, and γ-form of tellurene: Reprinted with permission from Ref. [27]. Copyright 2017, American Physical Society. Square-, chain-, and ring-Se: Reprinted with permission from Ref. [58]. Copyright 2017, IOP Publishing.

**Figure 3 molecules-28-00200-f003:**
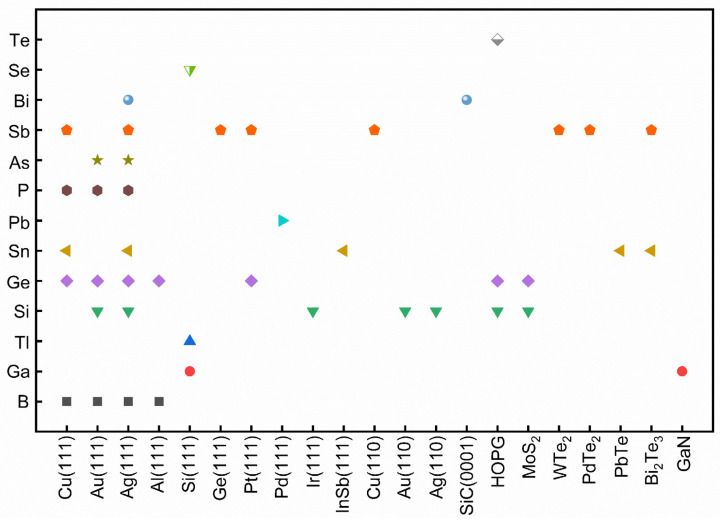
Summary of the successful substrate choices for the epitaxial growth of various non-graphene 2D Xenes. Each color represents one kind of Xenes.

**Figure 4 molecules-28-00200-f004:**
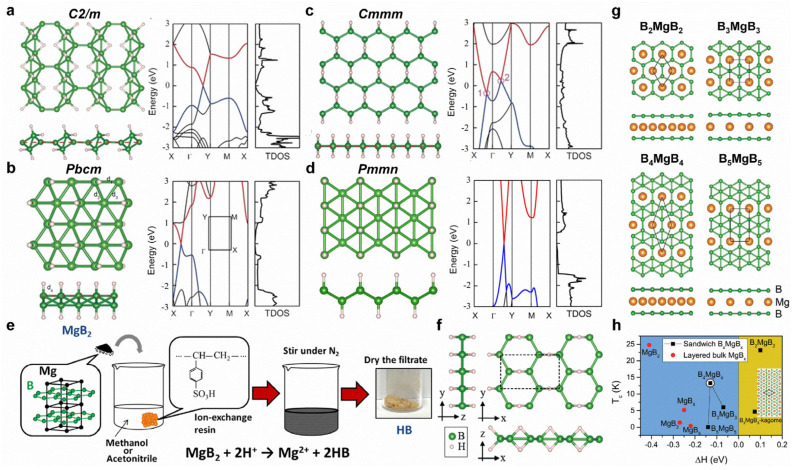
Surface modifications of borophene. (**a**–**d**) Top and side views as well as calculated band structures of (**a**) *C2/m*, (**b**) *Pbcm*, (**c**) *Cmmm*, and (**d**) *Pmmn* BH structures. The red and blue lines represent the unoccupied and occupied band dispersions near Fermi level, respectively. (**e**) Synthesis process of BH sheets. (**f**) Proposed structure model of synthesized BH sheet. (**g**) Top and side views of 2D sandwich structures of B_2_MgB_2_, B_3_MgB_3_, B_4_MgB_4_, and B_5_MgB_5_. (**h**) Calculated T_c_ as a function of the formation enthalpy. (**a**–**d**) Reprinted with permission from Ref. [107]. Copyright 2016, Wiley. (**e**,**f**) Reprinted with permission from Ref. [108]. Copyright 2017, American Chemical Society. (**g**,**h**) Reprinted with permission from Ref. [102]. Copyright 2017, Royal Society of Chemistry.

**Figure 5 molecules-28-00200-f005:**
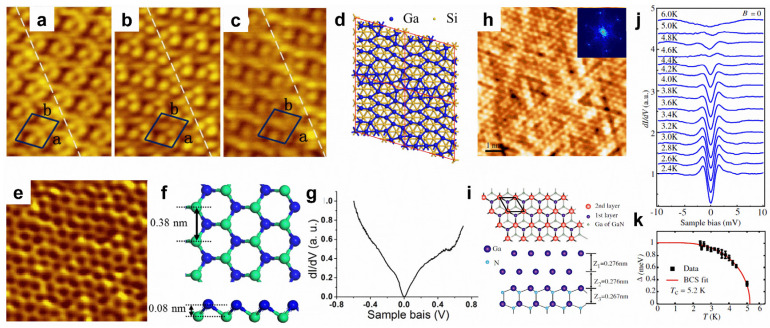
Surface modifications of gallenene. (**a**–**c**) STM images with biases of 2.0, 1.6, and 1.0 V of monolayer gallenene on Si(111) substrate. (**d**) Proposed configuration of monolayer gallenene on Si(111). (**e**) STM image of second-layer gallenene on Si(111) substrate. (**f**) Top and side views of the proposed structure for second-layer gallenene. (**g**) *dI/dV* spectrum recorded on the second-layer gallenene. (**h**) High-resolution STM image of gallenene on GaN(0001) substrate. (**i**) Top and side views of proposed gallenene structure on GaN(0001) substrate. (**j**) A series of *dI/dV* spectra recorded on gallenene/GaN at various temperatures. (**k**) Temperature-dependent superconducting gap magnitude and fitting to BCS gap function for gallenene/GaN. (**a**–**g**) Reprinted with permission from Ref. [110]. Copyright 2018, IOP Publishing Ltd. (**h**–**k**) Reprinted with permission from Ref. [111]. Copyright 2015, American Physical Society.

**Figure 6 molecules-28-00200-f006:**
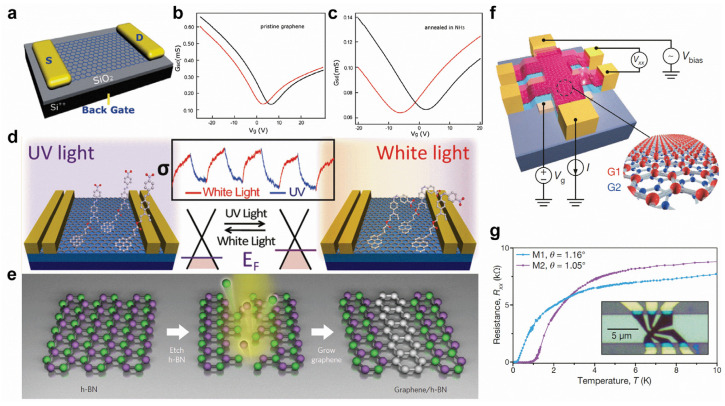
Surface modifications of graphene. (**a**–**c**) Comparison between the transfer characteristics of different graphene-based FETs (black and red curves were measured in air and in vacuum respectively). (**a**) Scheme of a graphene-based FET device. (**b**,**c**) The transport properties of (**b**) pristine graphene and (**c**) graphene annealed in NH_3_ after irradiation measured at V_sd_ = 0.03 V. (**d**) Light-driven conductance modulation of graphene with adsorption of DR1P molecules. (**e**) Fabrication of in-plane graphene/h-BN heterostructures. (**f**) Schematic of a twisted bilayer graphene device and four-probe measurement system. (**g**) Superconductivity in twisted bilayer graphene. (**a**–**c**) Reprinted with permission from Ref. [118]. Copyright, 2010 American Chemical Society. (**d**) Reprinted with permission from Ref. [36]. Copyright 2012, American Chemical Society. (**e**) Reprinted with permission from Ref. [42]. Copyright 2013, Springer Nature Ltd. (**f**,**g**) Reprinted with permission from Ref. [52]. Copyright 2018, Springer Nature Ltd.

**Figure 7 molecules-28-00200-f007:**
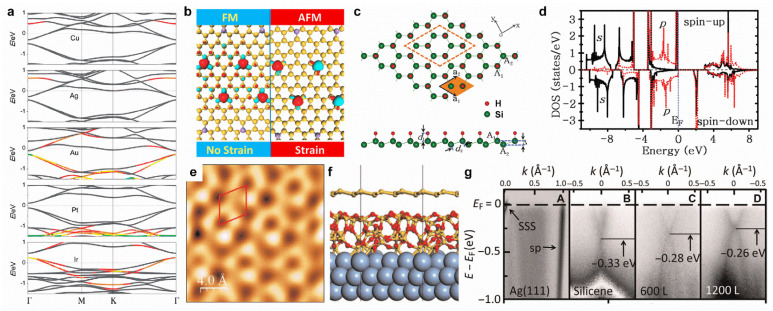
Surface modifications of silicene. (**a**) Electronic band structures of Cu-, Ag-, Au-, Pt-, and Ir-covered silicene with a coverage of 5.6%. The Fermi level is set to zero. (**b**) Strain-controlled magnetic exchange coupling in Mn-doped silicene. (**c**) Top and side views of the atomic structure for one-side semi-hydrogenated silicene. (**d**) Partial density of states calculated by HSE06 for semi-hydrogenated silicene. (**e**) High-resolution STM image of oxygen-intercalated epitaxial silicene grown on Ag(111). (**f**) Atomic structure of silicene/SiO_x_/Ag(111) from ab initio molecular dynamics (AIMD) simulation. (**g**) Energy versus k dispersion measured using ARPES for clean Ag(111) surface (A), as-grown √3 × √3 silicene formed on buffer layer (B), oxygen-intercalated silicene with an oxygen dose of 600 L (C), and intercalated silicene with an oxygen dose of 1200 L (D). (**a**) Reprinted with permission from Ref. [135]. Copyright 2014, Royal Society of Chemistry. (**b**) Reprinted with permission from Ref. [137]. Copyright 2017, American Chemical Society. (**c**,**d**) Reprinted with permission from Ref. [138]. Copyright 2012, Royal Society of Chemistry. (**e**–**g**) Reprinted with permission from Ref. [139]. Copyright 2016, American Association for the Advancement of Science.

**Figure 8 molecules-28-00200-f008:**
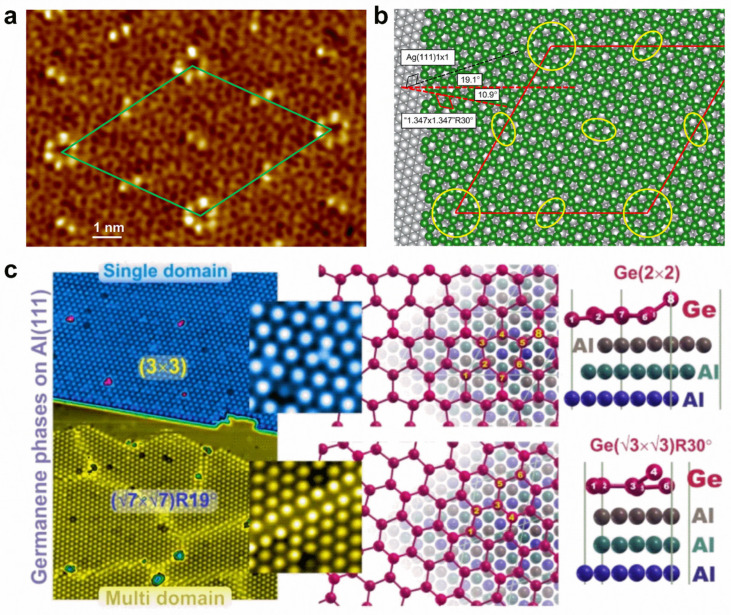
Surface modifications of germanene. (**a**) Atomic-scale STM image of germanene grown on a Ag(111) thin film. (**b**) Structural model of germanene on Ag(111). (**c**) Two different phases of germanene grown on an Al(111) surface. (**a**,**b**) Reprinted with permission from Ref. [151]. Copyright 2018, American Chemical Society. (**c**) Reprinted with permission from Ref. [152]. Copyright 2019, Springer Nature Ltd.

**Figure 9 molecules-28-00200-f009:**
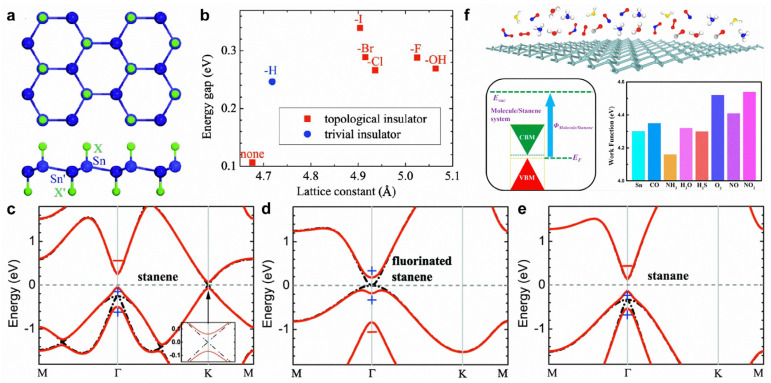
Surface modifications of stanene. (**a**) Crystal structure of decorated stanene by hydrogen or halogen atoms on both sides. (**b**) The calculated energy gaps of stanene and decorated stanene. (**c**–**e**) Band structures for (**c**) stanene, (**d**) fluorinated stanene, and (**e**) stanene with (red solid lines) and without (black dash-dotted lines) SOC. Parities of the Bloch states at the Γ point are denoted by +, −. (**f**) Structural model of stanene for the adsorption of various molecules and work function of stanene with various adsorbed molecules. (**a**–**e**) Reprinted with permission from Ref. [155]. Copyright 2013, American Physical Society. (**f**) Reprinted with permission from Ref. [157]. Copyright 2016, American Chemical Society.

**Figure 10 molecules-28-00200-f010:**
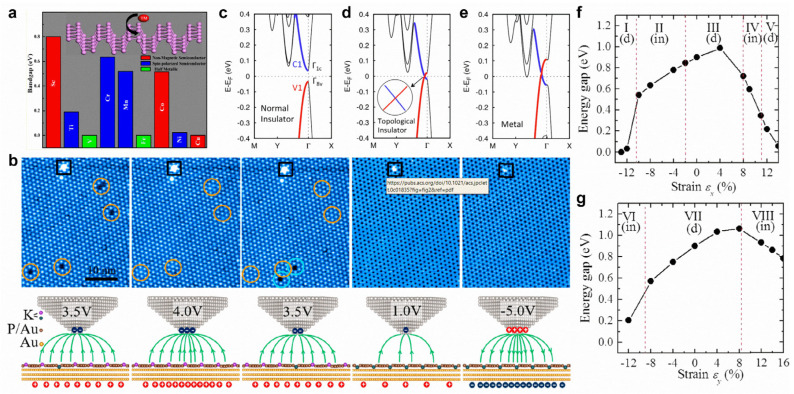
Surface modifications of phosphorene. (**a**) Electronic and magnetic properties of transition-metal-doped phosphorene. (**b**) STM images of blue phosphorene–Au network with 1 ML K deposition scanned with different bias voltages. (**c**–**e**) Band structures of 4-layer phosphorene with an external electric field of (**c**) 0 V Å^−1^, (**d**) 0.45 V Å^−1^, and (**e**) 0.6 V Å^−1^. (**f**,**g**) Bandgap of phosphorene as a function of strain: (**f**) ε_x_, applied in the zigzag and (**g**) ε_y_, in the armchair directions. (**a**) Reprinted with permission from Ref. [168]. Copyright 2015, American Chemical Society. (**b**) Reprinted with permission from Ref. [173]. Copyright 2020, American Chemical Society. (**c**–**e**) Reprinted with permission from Ref. [172]. Copyright 2015, American Chemical Society. (**f**,**g**) Reprinted with permission from Ref. [51]. Copyright 2014, American Physical Society.

**Figure 11 molecules-28-00200-f011:**
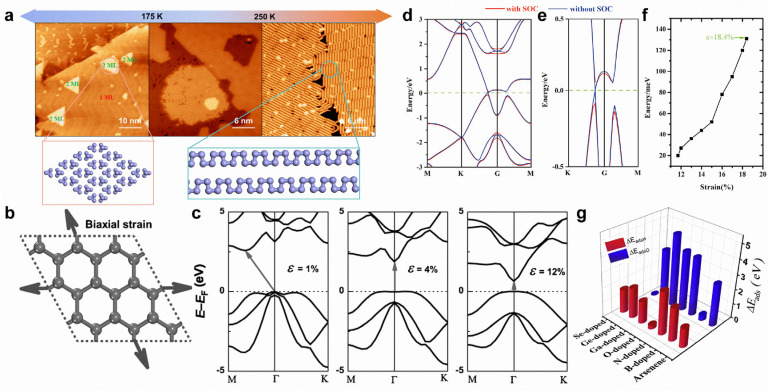
Surface modifications of arsenene. (**a**) Schematic diagram of substrate-temperature-dependent arsenic nanostructures on Au(111) substrate. (**b**) Schematic representation of arsenene under biaxial tensile strain. (**c**) Variations in electronic band structures of arsenene under various biaxial strains. The arrows represent the transitions of the bandgaps. (**d**,**e**) Band structure of arsenene under a biaxial tensile strain of 18.4% and an enlarged view. (**f**) Bandgap variation in arsenene with the tensile strain. (**g**) Adsorption energy of hydrogen and oxygen on pristine and B-, N-, O-, Ga-, Ge-, and Se-doped arsenene. (**a**) Reprinted with permission from Ref. [32]. Copyright 2022, American Chemical Society. (**b**,**c**) Reprinted with permission from Ref. [24]. Copyright 2015, Wiley. (**d**–**f**) Reprinted with permission from Ref. [177]. Copyright 2015, Royal Society of Chemistry. (**g**) Reprinted with permission from Ref. [181]. Copyright 2018, Elsevier Ltd.

**Figure 12 molecules-28-00200-f012:**
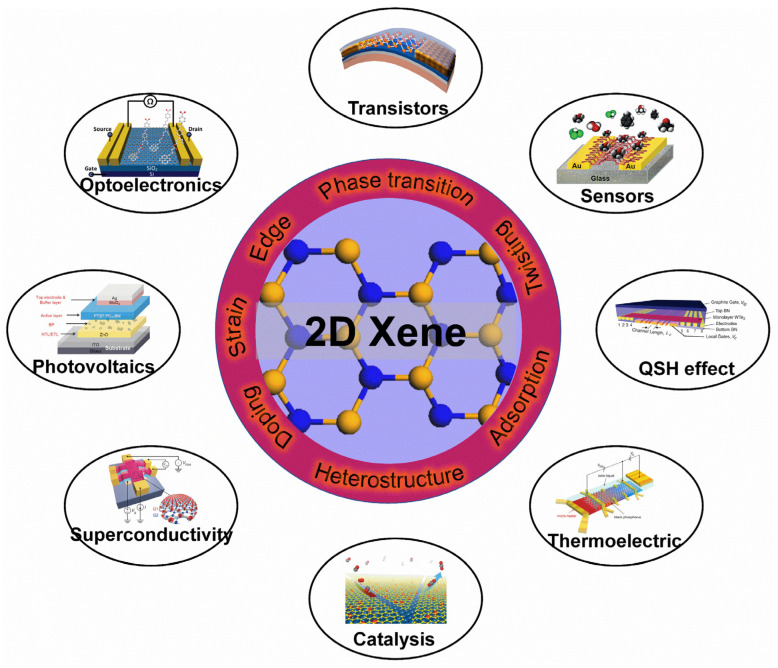
Applications of 2D Xenes with surface modifications. Transistors: Reprinted with permission from Ref. [207]. Copyright 2015, American Chemical Society. Optoelectronics: Reprinted with permission from Ref. [36]. Copyright 2012, American Chemical Society. Sensors: Reprinted with permission from Ref. [200]. Copyright 2015, American Chemical Society. Photovoltaics: Reprinted with permission from Ref. [199]. Copyright 2016, Wiley. QSH effect: Reprinted with permission from Ref. [208]. Copyright 2018, American Association for the Advancement of Science. Superconductivity: Reprinted with permission from Ref. [52]. Copyright 2018, Springer Nature Ltd. Thermoelectric: Reprinted with permission from Ref. [201]. Copyright 2016, American Chemical Society. Catalysis: Reprinted with permission from Ref. [202]. Copyright 2014, National Academy of Science.

## Data Availability

Data are contained within the article.

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
