# Peer review of "Recent Advances in Surface Modifications of Elemental Two-Dimensional Materials: Structures, Properties, and Applications"

_molecules, 2022, doi:10.3390/molecules28010200_

Round 1

Reviewer 1 Report

Chen and coauthors reviewed the recent advances in the elementary two-dimensional (2D) materials, mainly focusing on the tuning effects of surface modifications on the electronic, optical, mechanical and chemical properties of all 14 different Xenes discovered unit now. Since the advent of graphene in 2004, the field of elementary 2D materials has been developed into a fast growing direction, which suggests this review is quite timely. The manuscript is very detailed and comprehensive, and well organized, which is important for readers to quickly garner the status and challenges of this field. I recommend the acceptance of this paper.

Minor revision suggestions to authors:

1.      In line 219 on page 7, N-doping can not only induce n-type doping effect to graphene lattice, but can also give rise to p-type doping, depending on the configurations of the N substitutions. For example, graphitic N induces n-type doping, but pyridinic N induces p-type doping, as reported in Ma et al, Nano Lett. 18, 386–394 (2018).

2.      In line 224 on page 7, there is missing a blank in “graphene(Figure 6d)”.

3.      In line 403 on page 12, “up-down” should be “top-down”.

Reviewer 2 Report

Journal: Molecules

Ms. ID.: molecules-2123148

Title: Recent advances in surface modifications of elemental two-dimensional materials: structures, properties, and applications

Chen et al. reviewed the 2D materials’ structure, properties and applications. Focusing on elemental 2D materials, this review mainly summarizes the recently reported work about tuning the electronic, optical, mechanical, or chemical properties of Xenes via surface modifications achieved by controllable approaches (doping, adsorption, strain, intercalation, phase transition, etc.) to broaden the applications in various fields, including spintronics, electronics, optoelectronics, superconducting, photovoltaics, sensors, catalysis, and biomedicines. These advances in surface modification of Xenes have laid a theoretical and experimental foundation for developing 2D materials and their practical applications in diverse fields.

The manuscript is very well prepared. I find it suitable for the Journal. Still, I have some suggestions for improvement. I find Section 4 (Applications of 2D Xenens) too short. It should be given in more detail.
